# Hierarchical Representations for Efficient Architecture Search

**Hanxiao Liu**[*]
Carnegie Mellon University
`hanxiaol@cs.cmu.edu`

**Karen Simonyan, Oriol Vinyals, Chrisantha Fernando, Koray Kavukcuoglu**
DeepMind
`{simonyan,vinyals,chrisantha,korayk}@google.com`

## Abstract

We explore efficient neural architecture search methods and show that a simple yet powerful evolutionary algorithm can discover new architectures with excellent performance. Our approach combines a novel hierarchical genetic representation scheme that imitates the modularized design pattern commonly adopted by human experts, and an expressive search space that supports complex topologies. Our algorithm efficiently discovers architectures that outperform a large number of manually designed models for image classification, obtaining top-1 error of 3.6% on CIFAR-10 and 20.3% when transferred to ImageNet, which is competitive with the best existing neural architecture search approaches. We also present results using random search, achieving 0.3% less top-1 accuracy on CIFAR-10 and 0.1% less on ImageNet whilst reducing the search time from 36 hours down to 1 hour.

## 1 Introduction

Discovering high-performance neural network architectures required years of extensive research by human experts through trial and error. As far as the image classification task is concerned, state-of-the-art convolutional neural networks are going beyond deep, chain-structured layout (Simonyan & Zisserman, 2014; He et al., 2016a) towards increasingly more complex, graph-structured topologies (Szegedy et al., 2015; 2016; 2017; Larsson et al., 2016; Xie et al., 2016; Huang et al., 2016). The combinatorial explosion in the design space makes handcrafted architectures not only expensive to obtain, but also likely to be suboptimal in performance.

Recently, there has been a surge of interest in using algorithms to automate the manual process of architecture design. Their goal can be described as finding the optimal architecture in a given search space such that the validation accuracy is maximized on the given task. Representative architecture search algorithms can be categorized as random with weights prediction (Brock et al., 2017), Monte Carlo Tree Search (Negrinho & Gordon, 2017), evolution (Stanley & Miikkulainen, 2002; Xie & Yuille, 2017; Miikkulainen et al., 2017; Real et al., 2017), and reinforcement learning (Baker et al., 2016; Zoph & Le, 2016; Zoph et al., 2017; Zhong et al., 2017), among which reinforcement learning approaches have demonstrated the strongest empirical performance so far.

Architecture search can be computationally very intensive as each evaluation typically requires training a neural network. Therefore, it is common to restrict the search space to reduce complexity and increase efficiency of architecture search. Various constraints that have been used include: growing a convolutional "backbone" with skip connections (Real et al., 2017), a linear sequence of filter banks (Brock et al., 2017), or a directed graph where every node has exactly two predecessors (Zoph et al., 2017). In this work we constrain the search space by imposing a hierarchical network structure, while allowing flexible network topologies (directed acyclic graphs) at each level of the hierarchy. Starting from a small set of primitives such as convolutional and pooling operations at

---

[*]Work completed at DeepMind.

the bottom level of the hierarchy, higher-level computation graphs, or motifs, are formed by using lower-level motifs as their building blocks. The motifs at the top of the hierarchy are stacked multiple times to form the final neural network. This approach enables search algorithms to implement powerful hierarchical modules where any change in the motifs is propagated across the whole network immediately. This is analogous to the modularized design patterns used in many hand-crafted architectures, e.g. VGGNet (Simonyan & Zisserman, 2014), ResNet (He et al., 2016a), and Inception (Szegedy et al., 2016) are all comprised of building blocks. In our case, a hierarchical architecture is discovered through evolutionary or random search.

The evolution of neural architectures was studied as a sub-task of *neuroevolution* (Holland, 1975; Miller et al., 1989; Yao, 1999; Stanley & Miikkulainen, 2002; Floreano et al., 2008), where the topology of a neural network is simultaneously evolved along with its weights and hyperparameters. The benefits of indirect encoding schemes, such as multi-scale representations, have historically been discussed in Gruau et al. (1994); Kitano (1990); Stanley (2007); Stanley et al. (2009). Despite these pioneer studies, evolutionary or random architecture search has not been investigated at larger scale on image classification benchmarks until recently (Real et al., 2017; Miikkulainen et al., 2017; Xie & Yuille, 2017; Brock et al., 2017; Negrinho & Gordon, 2017). Our work shows that the power of simple search methods can be substantially enhanced using well-designed search spaces.

Our experimental setup resembles Zoph et al. (2017), where an architecture found using reinforcement learning obtained the state-of-the-art performance on ImageNet. Our work reveals that random or evolutionary methods, which so far have been seen as less efficient, can scale and achieve competitive performance on this task if combined with a powerful architecture representation, whilst utilizing significantly less computational resources.

To summarize, our main contributions are:

1. We introduce hierarchical representations for describing neural network architectures.
2. We show that competitive architectures for image classification can be obtained even with simplistic random search, which demonstrates the importance of search space construction.
3. We present a scalable variant of evolutionary search which further improves the results and achieves the best published results[1] among evolutionary architecture search techniques.

## 2 ARCHITECTURE REPRESENTATIONS

We first describe flat representations of neural architectures (Sect. 2.1), where each architecture is represented as a single directed acyclic graph of primitive operations. Then we move on to hierarchical representations (Sect. 2.2) where smaller graph motifs are used as building blocks to form larger motifs. Primitive operations are discussed in Sect. 2.3.

### 2.1 FLAT ARCHITECTURE REPRESENTATION

We consider a family of neural network architectures represented by a single-source, single-sink computation graph that transforms the input at the source to the output at the sink. Each node of the graph corresponds to a feature map, and each directed edge is associated with some primitive operation (e.g. convolution, pooling, etc.) that transforms the feature map in the input node and passes it to the output node.

Formally, an architecture is defined by the representation $(G, \boldsymbol{o})$, consisting of two ingredients:

1. A set of available operations $\boldsymbol{o} = \{o_1, o_2, \dots\}$.
2. An adjacency matrix $G$ specifying the neural network graph of operations, where $G_{ij} = k$ means that the $k$-th operation $o_k$ is to be placed between nodes $i$ and $j$.

The architecture is obtained by assembling operations $\boldsymbol{o}$ according to the adjacency matrix $G$:

$$arch = assemble(G, \boldsymbol{o}) \tag{1}$$

---

[1]at the moment of paper submission; see Real et al. (2018) for a more recent study of evolutionary methods for architecture search.

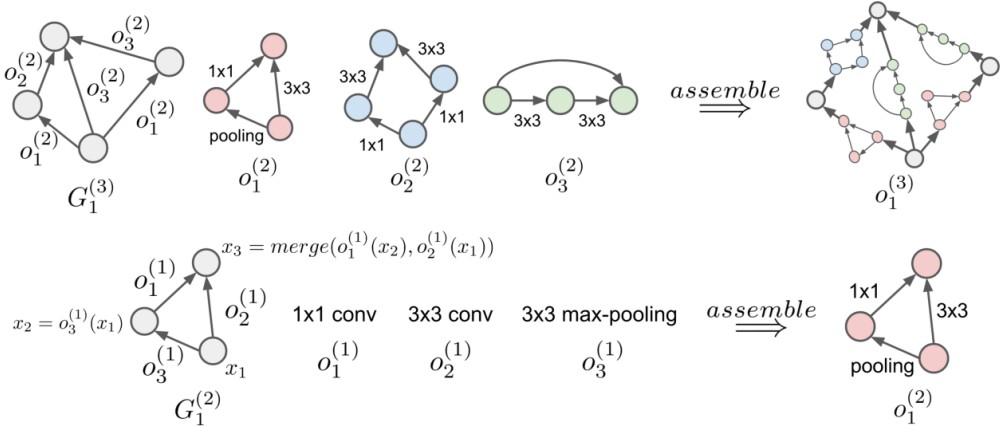

Figure 1: An example of a three-level hierarchical architecture representation. The bottom row shows how level-1 primitive operations $o_1^{(1)}, o_2^{(1)}, o_3^{(1)}$ are assembled into a level-2 motif $o_1^{(2)}$. The top row shows how level-2 motifs $o_1^{(2)}, o_2^{(2)}, o_3^{(2)}$ are then assembled into a level-3 motif $o_1^{(3)}$.

in a way that the resulting neural network sequentially computes the feature map $x_i$ of each node $i$ from the feature maps $x_j$ of its predecessor nodes $j$ following the topological ordering:

$$x_i = merge\left[\{o_{G_{ij}}(x_j)\}_{j<i}\right], \quad i = 2, \ldots, |G| \tag{2}$$

Here, $|G|$ is the number of nodes in a graph, and $merge$ is an operation combining multiple feature maps into one, which in our experiments was implemented as depthwise concatenation. An alternative option of element-wise addition is less flexible as it requires the incoming feature maps to contain the same number of channels, and is strictly subsumed by concatenation if the resulting $x_i$ is immediately followed by a $1 \times 1$ convolution.

## 2.2 HIERARCHICAL ARCHITECTURE REPRESENTATION

The key idea of the hierarchical architecture representation is to have several motifs at different levels of hierarchy, where lower-level motifs are used as building blocks (operations) during the construction of higher-level motifs.

Consider a hierarchy of $L$ levels where the $\ell$-th level contains $M_\ell$ motifs. The highest-level $\ell = L$ contains only a single motif corresponding to the full architecture, and the lowest level $\ell = 1$ is the set of primitive operations. We recursively define $o_m^{(\ell)}$, the $m$-th motif in level $\ell$, as the composition of lower-level motifs $\boldsymbol{o}^{(\ell-1)} = \{o_1^{(\ell-1)}, o_2^{(\ell-1)}, \ldots, o_{M_{(\ell-1)}}^{(\ell-1)}\}$ according to its network structure $G_m^{(\ell)}$:

$$o_m^{(\ell)} = assemble\left(G_m^{(\ell)}, \boldsymbol{o}^{(\ell-1)}\right), \quad \forall \ell = 2, \ldots, L \tag{3}$$

A hierarchical architecture representation is therefore defined by $\left(\{\{G_m^{(\ell)}\}_{m=1}^{M_\ell}\}_{\ell=2}^{L}, \boldsymbol{o}^{(1)}\right)$, as it is determined by network structures of motifs at all levels and the set of bottom-level primitives. The assembly process is illustrated in Fig. 1.

## 2.3 PRIMITIVE OPERATIONS

We consider the following six primitives at the bottom level of the hierarchy ($\ell = 1, M_\ell = 6$):

- $1 \times 1$ convolution of $C$ channels
- $3 \times 3$ depthwise convolution
- $3 \times 3$ separable convolution of $C$ channels
- $3 \times 3$ max-pooling

- $3 \times 3$ average-pooling
- identity

If applicable, all primitives are of stride one and the convolved feature maps are padded to preserve their spatial resolution. All convolutional operations are followed by batch normalization and ReLU activation (Ioffe & Szegedy, 2015); their number of channels is fixed to a constant $C$. We note that convolutions with larger receptive fields and more channels can be expressed as motifs of such primitives. Indeed, large receptive fields can be obtained by stacking $3 \times 3$ convolutions in a chain structure (Simonyan & Zisserman, 2014), and wider convolutions with more channels can be obtained by merging the outputs of multiple convolutions through depthwise concatenation.

We also introduce a special *none* op, which indicates that there is no edge between nodes $i$ and $j$. It is added to the pool of operations at each level.

# 3 EVOLUTIONARY ARCHITECTURE SEARCH

Evolutionary search over neural network architectures can be performed by treating the representations of Sect. 2 as genotypes. We first introduce an action space for mutating hierarchical genotypes (Sect. 3.1), as well as a diversification-based scheme to obtain the initial population (Sect. 3.2). We then describe tournament selection and random search in Sect. 3.3, and our distributed implementation in Sect. 3.4.

## 3.1 MUTATION

A single mutation of a hierarchical genotype consists of the following sequence of actions:

1. Sample a target non-primitive level $\ell \geq 2$.
2. Sample a target motif $m$ in the target level.
3. Sample a random successor node $i$ in the target motif.
4. Sample a random predecessor node $j$ in the target motif.
5. Replace the current operation $o_k^{(\ell-1)}$ between $j$ and $i$ with a randomly sampled operation $o_{k'}^{(\ell-1)}$.

In the case of flat genotypes which consist of two levels (one of which is the fixed level of primitives), the first step is omitted and $\ell$ is set to 2. The mutation can be summarized as:

$$[G_m^{(\ell)}]_{ij} = k' \tag{4}$$

where $\ell, m, i, j, k'$ are randomly sampled from uniform distributions over their respective domains. Notably, the above mutation process is powerful enough to perform various modifications on the target motif, such as:

1. **Add a new edge**: if $o_k^{(\ell-1)} = none$ and $o_{k'}^{(\ell-1)} \neq none$.
2. **Alter an existing edge**: if $o_k^{(\ell-1)} \neq none$ and $o_{k'}^{(\ell-1)} \neq none$ and $o_{k'}^{(\ell-1)} \neq o_k^{(\ell-1)}$.
3. **Remove an existing edge**: if $o_k^{(\ell-1)} \neq none$ and if $o_{k'}^{(\ell-1)} = none$.

## 3.2 INITIALIZATION

To initialize the population of genotypes, we use the following strategy:

1. Create a "trivial" genotype where each motif is set to a chain of identity mappings.
2. Diversify the genotype by applying a large number (e.g. 1000) of random mutations.

In contrast to several previous works where genotypes are initialized by trivial networks (Stanley & Miikkulainen, 2002; Real et al., 2017), the above diversification-based scheme not only offers a

---

**Algorithm 1:** ASYNCEVO Asynchronous Evolution (Controller)

---

**Input:** Data queue $\mathcal{Q}$ containing initial genotypes; Memory table $\mathcal{M}$ recording evaluated
      genotypes and their fitness.
**while** *True* **do**
    **if** HASIDLEWORKER() **then**
        $genotype \leftarrow$ ASYNCTOURNAMENTSELECT($\mathcal{M}$)
        $genotype' \leftarrow$ MUTATE($genotype$)
        $\mathcal{Q} \leftarrow \mathcal{Q} \cup genotype'$

---

**Algorithm 2:** ASYNCEVO Asynchronous Evolution (Worker)

---

**Input:** Training set $\mathcal{T}$, validation set $\mathcal{V}$; Shared memory table $\mathcal{M}$ and data queue $\mathcal{Q}$.
**while** *True* **do**
    **if** $|\mathcal{Q}| > 0$ **then**
        $genotype \leftarrow \mathcal{Q}.\text{pop}()$
        $arch \leftarrow$ ASSEMBLE($genotype$)
        $model \leftarrow$ TRAIN($arch, \mathcal{T}$)
        $fitness \leftarrow$ EVALUATE($model, \mathcal{V}$)
        $\mathcal{M} \leftarrow \mathcal{M} \cup (genotype, fitness)$

---

good initial coverage of the search space with non-trivial architectures, but also helps to avoid an additional bias introduced by handcrafted initialization routines. In fact, this strategy ensures initial architectures are reasonably well-performing even without any search, as suggested by our random sample results in Table 1.

### 3.3 SEARCH ALGORITHMS

Our evolutionary search algorithm is based on tournament selection (Goldberg & Deb, 1991). Starting from an initial population of random genotypes, tournament selection provides a mechanism to pick promising genotypes from the population, and to place its mutated offspring back into the population. By repeating this process, the quality of the population keeps being refined over time. We always train a model from scratch for a fixed number of iterations, and we refer to the training and evaluation of a single model as an evolution step. The genotype with the highest fitness (validation accuracy) among the entire population is selected as the final output after a fixed amount of time.

A tournament is formed by a random set of genotypes sampled from the current effective population, among which the individual with the highest fitness value wins the tournament. The selection pressure is controlled by the tournament size, which is set to $5\%$ of the population size in our case. We do not remove any genotypes from the population, allowing it to grow with time, maintaining architecture diversity. Our evolution algorithm is similar to the binary tournament selection used in a recent large-scale evolutionary method (Real et al., 2017).

We also investigated random search, a simpler strategy which has not been sufficiently explored in the literature, as an alternative to evolution. In this case, a population of genotypes is generated randomly, the fitness is computed for each genotype in the same way as done in evolution, and the genotype with the highest fitness is selected as the final output. The main advantage of this method is that it can be run in parallel over the entire population, substantially reducing the search time.

### 3.4 IMPLEMENTATION

Our distributed implementation is asynchronous, consisting of a single controller responsible for performing evolution over the genotypes, and a set of workers responsible for their evaluation. Both parties have access to a shared tabular memory $\mathcal{M}$ recording the population of genotypes and their fitness, as well as a data queue $\mathcal{Q}$ containing the genotypes with unknown fitness which should be evaluated.

Specifically, the controller will perform tournament selection of a genotype from $\mathcal{M}$ whenever a worker becomes available, followed by the mutation of the selected genotype and its insertion into $\mathcal{Q}$ for fitness evaluation (Algorithm 1). A worker will pick up an unevaluated genotype from $\mathcal{Q}$ whenever there is one available, assemble it into an architecture, carry out training and validation, and then record the validation accuracy (fitness) in $\mathcal{M}$ (Algorithm 2). Architectures are trained from scratch for a fixed number of steps with random weight initialization. We do not rely on weight inheritance as in (Real et al., 2017), though incorporating it into our system is possible. Note that during architecture evolution no synchronization is required, and all workers are fully occupied.

## 4 EXPERIMENTS AND RESULTS

### 4.1 EXPERIMENTAL SETUP

In our experiments, we use the proposed search framework to learn the architecture of a convolutional cell, rather than the entire model. The reason is that we would like to be able to quickly compute the fitness of the candidate architecture and then transfer it to a larger model, which is achieved by using less cells for fitness computation and more cells for full model evaluation. A similar approach has recently been used in (Zoph et al., 2017; Zhong et al., 2017).

Architecture search is carried out entirely on the CIFAR-10 training set, which we split into two sub-sets of 40K training and 10K validation images. Candidate models are trained on the training subset, and evaluated on the validation subset to obtain the fitness. Once the search process is over, the selected cell is plugged into a large model which is trained on the combination of training and validation sub-sets, and the accuracy is reported on the CIFAR-10 test set. We note that the test set is never used for model selection, and it is only used for final model evaluation. We also evaluate the cells, learned on CIFAR-10, in a large-scale setting on the ImageNet challenge dataset (Sect. 4.3).

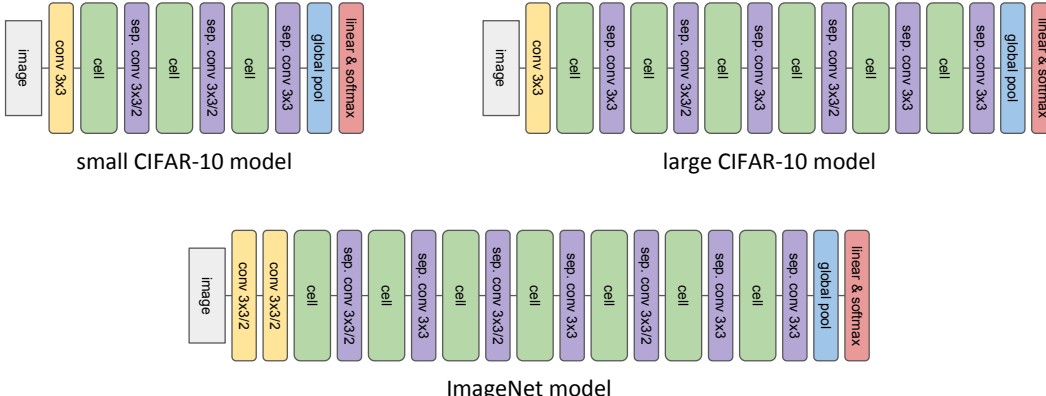

Figure 2: Image classification models constructed using the cells optimized with architecture search. *Top-left:* small model used during architecture search on CIFAR-10. *Top-right:* large CIFAR-10 model used for learned cell evaluation. *Bottom:* ImageNet model used for learned cell evaluation.

For CIFAR-10 experiments we use a model which consists of $3 \times 3$ convolution with $c_0$ channels, followed by 3 groups of learned convolutional cells, each group containing $N$ cells. After each cell (with $c$ input channels) we insert $3 \times 3$ separable convolution which has stride 2 and $2c$ channels if it is the last cell of the group, and stride 1 and $c$ channels otherwise. The purpose of these convolutions is to control the number of channels as well as reduce the spatial resolution. The last cell is followed by global average pooling and a linear softmax layer.

For fitness computation we use a smaller model with $c_0 = 16$ and $N = 1$, shown in Fig. 2 (top-left). It is trained using SGD with $0.9$ momentum for 5000 steps, starting with the learning rate $0.1$, which is reduced by 10x after 4000 and 4500 steps. The batch size is 256, and the weight decay value is $3 \cdot 10^{-4}$. We employ standard training data augmentation where a $24 \times 24$ crop is randomly sampled from a $32 \times 32$ image, followed by random horizontal flipping. The evaluation is performed on the full size $32 \times 32$ image.

**A note on variance.** We found that the variance due to optimization was non-negligible, and we believe that reporting it is important for performing a fair comparison and assessing model capabilities. When training CIFAR models, we have observed standard deviation of up to 0.2% using the exact same setup. The solution we adopted was to compute the fitness as the average accuracy over 4 training-evaluation runs.

For the evaluation of the learned cell architecture on CIFAR-10, we use a larger model with $c_0 = 64$ and $N = 2$, shown in Fig. 2 (top-right). The larger model is trained for 80K steps, starting with a learning rate 0.1, which is reduced by 10x after 40K, 60K, and 70K steps. The rest of the training settings are the same as used for fitness computation. We report mean and standard deviation computed over 5 training-evaluation runs.

For the evaluation on the ILSVRC ImageNet challenge dataset (Russakovsky et al., 2015), we use an architecture similar to the one used for CIFAR, with the following changes. An input $299 \times 299$ image is passed through two convolutional layers with 32 and 64 channels and stride 2 each. It is followed by 4 groups of convolutional cells where the first group contains a single cell (and has $c_0 = 64$ input channels), and the remaining three groups have $N = 2$ cells each (Fig. 2, bottom). We use SGD with momentum which is run for 200K steps, starting with a learning rate of 0.1, which is reduced by 10x after 100K, 150K, and 175K steps. The batch size is 1024, and weight decay is $10^{-4}$. We did not use auxiliary losses, weight averaging, label smoothing or path dropout empirically found effective in (Zoph et al., 2017). The training augmentation is the same as in (Szegedy et al., 2016), and consists in random crops, horizontal flips and brightness and contrast changes. We report the single-crop top-1 and top-5 error on the ILSVRC validation set.

## 4.2 Architecture search on CIFAR-10

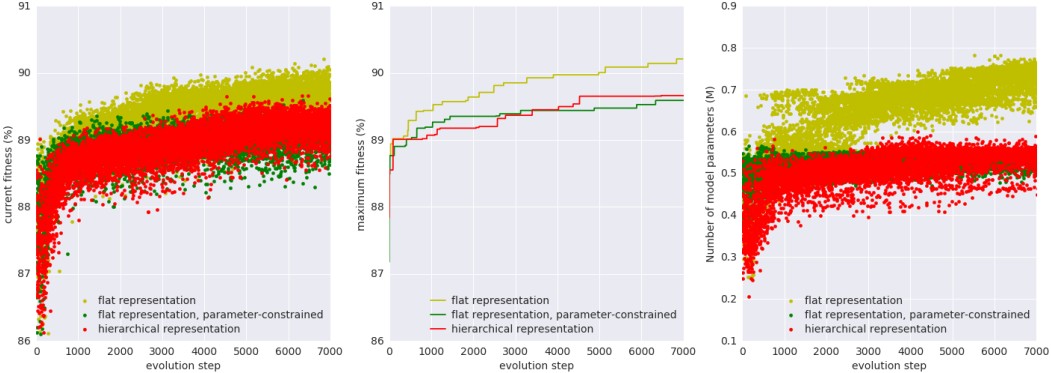

Figure 3: Fitness and number of parameters vs evolution step for flat and hierarchical representations. *Left:* fitness of a genotype generated at each evolution step. *Middle:* maximum fitness across all genotypes generated before each evolution step. *Right:* number of parameters in the small CIFAR-10 model constructed using the genotype generated at each evolution step.

We run the evolution on flat and hierarchical genotypes for 7000 steps using 200 GPU workers. The initial size of the randomly initialized population is 200, which later grows as a result of tournament selection and mutation (Sect. 3). For the hierarchical representation, we use three levels ($L = 3$), with $M_1 = 6, M_2 = 6, M_3 = 1$. Each of the level-2 motifs is a graph with $|G^{(2)}| = 4$ nodes, and the level-3 motif is a graph with $|G^{(3)}| = 5$ nodes. Each level-2 motif is followed by a $1 \times 1$ convolution with the same number of channels as on the motif input to reduce the number of parameters. For the flat representation, we used a graph with 11 nodes to achieve a comparable number of edges.

The evolution process is visualized in Fig. 3. The left plot shows the fitness of the genotype generated at each step of evolution: the fitness grows fast initially, and plateaus over time. The middle plot shows the best fitness observed by each evolution step. Since the first 200 steps correspond to a random initialization and mutation starts after that, the best architecture found at step 200 corresponds to the output of random search over 200 architectures.

Fig. 3 (right) shows the number of parameters in the small network (used for fitness computation), constructed using the genotype produced at each step. Notably, flat genotypes achieve higher fitness, but at the cost of larger parameter count. We thus also consider a parameter-constrained variant of the flat genotype, where only the genotypes with the number of parameters under a fixed threshold are permitted; the threshold is chosen so that the flat genotype has a similar number of parameters to the hierarchical one. In this setting hierarchical and flat genotypes achieve similar fitness.

To demonstrate that improvement in fitness of the hierarchical architecture is correlated with the improvement in the accuracy of the corresponding large model trained till convergence, we plot the relative accuracy improvements in Fig. 4.

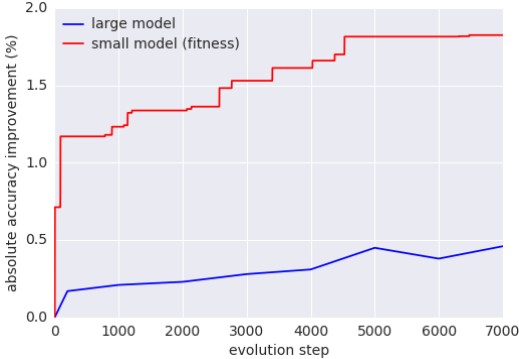

Figure 4: Accuracy improvement over the course of evolution, measured with respect to the first random genotype. The small model is the model used for fitness computation during evolution (its absolute fitness value is shown with the red curve in Fig. 3 (middle)). The large model is the model where the evolved cell architecture is deployed for training and evaluation.

As far as the architecture search time is concerned, it takes 1 hour to compute the fitness of one architecture on a single P100 GPU (which involves 4 rounds of training and evaluation). Using 200 GPUs, it thus takes 1 hour to perform random search over 200 architectures and 1.5 days to do the evolutionary search with 7000 steps. This is significantly faster than 11 days using 250 GPUs reported by (Real et al., 2017) and 4 days using 450 GPUs reported by (Zoph et al., 2017).

| Search Method | CIFAR-10 error (%) | ImageNet Top-1/Top-5 error (%) |
|---|---|---|
| Flat repr-n, random architecture | $4.56 \pm 0.11$ | 21.4/5.8 |
| Flat repr-n, random search (200 samples) | $4.02 \pm 0.11$ | 20.8/5.7 |
| Flat repr-n, evolution (7000 samples) | $3.92 \pm 0.06$ | 20.6/5.6 |
| Flat repr-n, parameter-constrained, evolution (7000 samples) | $4.17 \pm 0.08$ | 21.2/5.8 |
| Hier. repr-n, random architecture | $4.21 \pm 0.11$ | 21.5/5.8 |
| Hier. repr-n, random search (200 samples) | $4.04 \pm 0.2$ | 20.4/5.3 |
| Hier. repr-n, random search (7000 samples) | $3.91 \pm 0.15$ | 21.0/5.5 |
| Hier. repr-n, evolution (7000 samples) | $\mathbf{3.75 \pm 0.12}$ | **20.3/5.2** |

Table 1: Classification results on the CIFAR-10 test set and ILSVRC validation set obtained using the architectures found using various representations and search methods.

## 4.3 ARCHITECTURE EVALUATION ON CIFAR-10 AND IMAGENET

We now turn to the evaluation of architectures found using random and evolutionary search on CIFAR-10 and ImageNet. The results are presented in Table 1.

First, we note that randomly sampled architectures already perform surprisingly well, which we attribute to the representation power of our architecture spaces. Second, random search over 200 architectures achieves very competitive results on both CIFAR-10 and ImageNet, which is remarkable considering it took 1 hour to carry out. This demonstrates that well-constructed architecture repre-

sentations, coupled with diversified sampling and simple search form a simple but strong baseline for architecture search. Our best results are achieved using evolution over hierarchical representations: $3.75\% \pm 0.12\%$ classification error on the CIFAR-10 test set (using $c_0 = 64$ channels), which is further improved to $3.63\% \pm 0.10\%$ with more channels ($c_0 = 128$). On the ImageNet validation set, we achieve 20.3% top-1 classification error and 5.2% top-5 error. We put these results in the context of the state of the art in Tables 2 and 3. We achieve the best published results on CIFAR-10 using evolutionary architecture search, and also demonstrate competitive performance compared to the best published methods on both CIFAR-10 and ImageNet. Our ImageNet model has 64M parameters, which is comparable to Inception-ResNet-v2 (55.8M) but larger than NASNet-A (22.6M).

| Model | Error (%) |
|---|---|
| ResNet-1001 + pre-activation (He et al., 2016b) | 4.62 |
| Wide ResNet-40-10 + dropout (Zagoruyko & Komodakis, 2016) | 3.8 |
| DenseNet (k=24) (Huang et al., 2016) | 3.74 |
| DenseNet-BC (k=40) (Huang et al., 2016) | 3.46 |
| MetaQNN (Baker et al., 2016) | 6.92 |
| NAS v3 (Zoph & Le, 2016) | 3.65 |
| Block-QNN-A (Zhong et al., 2017) | 3.60 |
| NASNet-A (Zoph et al., 2017) | 3.41 |
| Evolving DNN (Miikkulainen et al., 2017) | 7.3 |
| Genetic CNN (Xie & Yuille, 2017) | 7.10 |
| Large-scale Evolution (Real et al., 2017) | 5.4 |
| SMASH (Brock et al., 2017) | 4.03 |
| Evolutionary search, hier. repr., $c_0 = 64$ | $3.75 \pm 0.12$ |
| Evolutionary search, hier. repr., $c_0 = 128$ | $3.63 \pm 0.10$ |

Table 2: Classification error on the CIFAR-10 test set obtained using state-of-the-art models as well as the best-performing architecture found using the proposed architecture search framework. Existing models are grouped as (from top to bottom): handcrafted architectures, architectures found using reinforcement learning, and architectures found using random or evolutionary search.

| Model | Top-1 error (%) | Top-5 error (%) |
|---|---|---|
| Inception-v3 (Szegedy et al., 2016) | 21.2 | 5.6 |
| Xception (Chollet, 2016) | 21.0 | 5.5 |
| Inception-ResNet-v2 (Szegedy et al., 2017) | 19.9 | 4.9 |
| NASNet-A (Zoph et al., 2017) | 19.2 | 4.7 |
| Evolutionary search, hier. repr., $c_0 = 64$ | 20.3 | 5.2 |

Table 3: Classification error on the ImageNet validation set obtained using state-of-the-art models as well as the best-performing architecture found using our framework.

The evolved hierarchical cell is visualized in Appendix A, which shows that architecture search have discovered a number of skip connections. For example, the cell contains a direct skip connection between input and output: nodes 1 and 5 are connected by Motif 4, which in turn contains a direct connection between input and output. The cell also contains several internal skip connections, through Motif 5 (which again comes with an input-to-output skip connection similar to Motif 4).

## 5 CONCLUSION

We have presented an efficient evolutionary method that identifies high-performing neural architectures based on a novel hierarchical representation scheme, where smaller operations are used as the building blocks to form the larger ones. Notably, we show that strong results can be obtained even using simplistic search algorithms, such as evolution or random search, when coupled with a well-designed architecture representation. Our best architecture yields the state-of-the-art result on

CIFAR-10 among evolutionary methods and successfully scales to ImageNet with highly competitive performance.

## ACKNOWLEDGEMENTS

The authors thank Jacob Menick, Pushmeet Kohli, Yujia Li, Simon Osindero, and many other colleagues at DeepMind for helpful comments and discussions.

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

## A   ARCHITECTURE VISUALIZATION

Visualization of the learned cell and motifs of our best-performing hierarchical architecture. Note that only motifs 1,3,4,5 are used to construct the cell, among which motifs 3 and 5 are dominating.

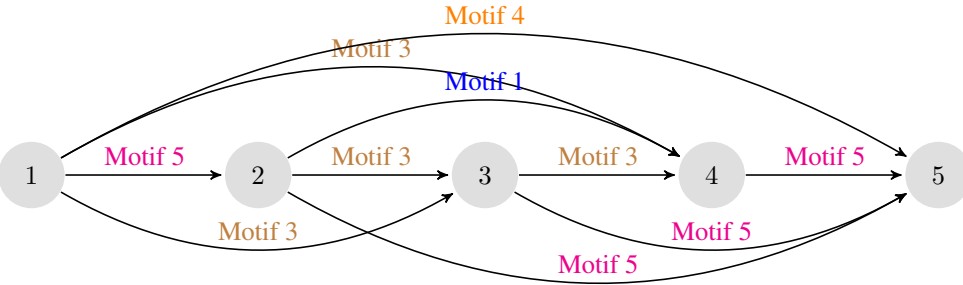

Figure 5: Cell

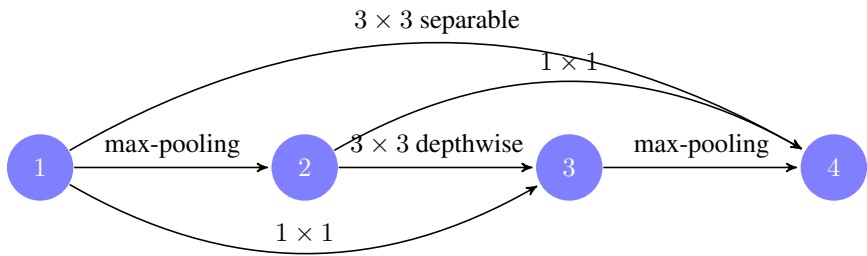

Figure 6: Motif 1

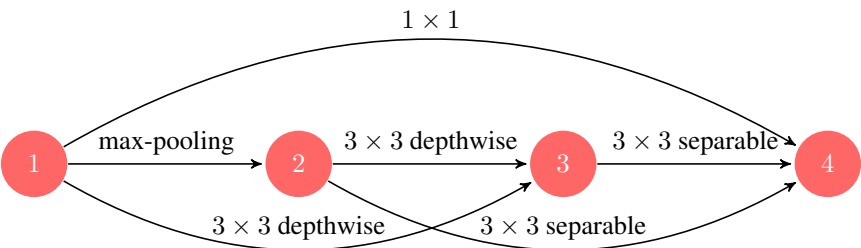

Figure 7: Motif 2

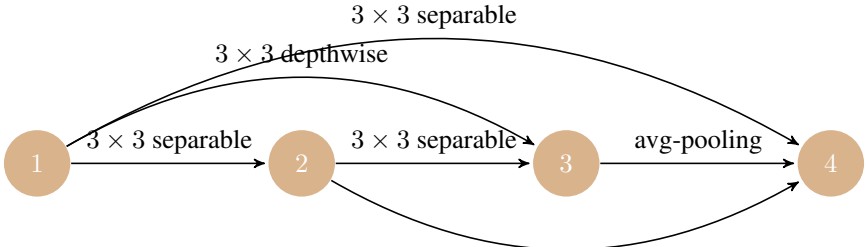

Figure 8: Motif 3

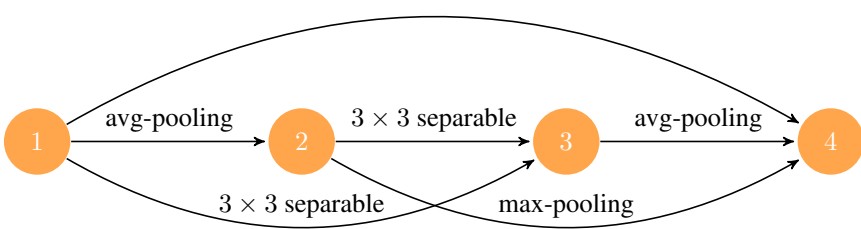

Figure 9: Motif 4

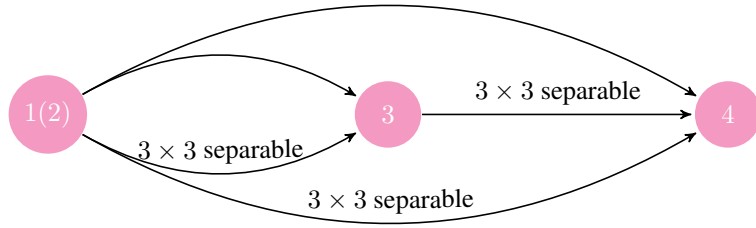

Figure 10: Motif 5

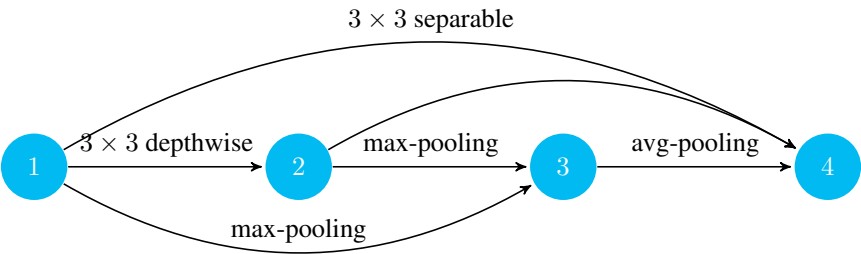

Figure 11: Motif 6

