# OpenReview forum: "Hierarchical Representations for Efficient Architecture Search"
_ICLR.cc/2018/Conference — Accept (Poster)_

### Official Review · AnonReviewer2 · 2017-11-26
**In this paper, the authors propose a novel evolutionary algorithm for neural architecture search.**

**Rating:** 6
**Confidence:** 3

**Review:**

The fundamental contribution of the article is the explicit use of compositionality in the definition of the search space. Instead of merely defining an architecture as a Directed Acyclic Graph (DAG), with nodes corresponding to feature maps and edges to primitive operations, the approach in this paper introduces a hierarchy of architectures of this form. Each level of the hierarchy utilises the existing architectures in the preceding level as candidate operations to be applied in the edges of the DAG. As a result, this would allow the evolutionary search algorithm to design modules which might be then reused in different edges of the DAG corresponding to the final architecture, which is located at the top level in the hierarchy.

Manually designing novel neural architectures is a laborious, time-consuming process. Therefore, exploring new approaches to automatise this task is a problem of great relevance for the field.

Overall, the paper is well-written, clear in its exposition and technically sound. While some hyperparameter and design choices could perhaps have been justified in greater detail, the paper is mostly self-contained and provides enough information to be reproducible.

The fundamental contribution of this article, when put into the context of the many recent publications on the topic of automatic neural architecture search, is the introduction of a hierarchy of architectures as a way to build the search space. Compared to existing work, this approach should emphasise modularity, making it easier for the evolutionary search algorithm to discover architectures that extensively reuse simpler blocks as part of the model. Exploiting compositionality in model design is not novel per se (e.g. [1,2]), but it is to the best of my knowledge the first explicit application of this idea in neural architecture search.

Nevertheless, while the idea behind the proposed approach is definitely interesting, I believe that the experimental results do not provide sufficiently compelling evidence that the resulting method substantially outperforms the non-hierarchical, flat representation of architectures used in other publications. In particular, the results highlighted in Figure 3 and Table 1 seem to indicate that the difference in performance between both paradigms is rather small. Moreover, the performance gap between the flat and hierarchical representations of the search space, as reported in Table 1, remains smaller than the performance gap between the best performing of the approaches proposed in this article and NASNet-A (Zoph et al., 2017), as reported in Tables 2 and 3.

Another concern I have is regarding the definition of the mutation operators in Section 3.1. While not explicitly stated, I assume that all sampling steps are performed uniformly at random (otherwise please clarify it). If that was indeed the case, there is a systematic asymmetry between the probability to add and remove an edge, making the former considerably more likely. This could bias the architectures towards fully-connected DAGs, as indeed seems to occur based on the motifs reported in Appendix A.

Finally, while the main motivation behind neural architecture search is to automatise the design of new models, the approach here presented introduces a non-negligible number of hyperparameters that could potentially have a considerable impact and need to be selected somehow. This includes, for instance, the number of levels in the hierarchy (L), the number of motifs at each level in the hierarchy (M_l), the number of nodes in each graph at each level in the hierarchy (| G^{(l)} |), as well as the set of primitive operations. I believe the paper would be substantially strengthened if the authors explored how robust the resulting approach is with respect to perturbations of these hyperparameters, and/or provided users with a principled approach to select reasonable values.

References:

[1] Grosse, Roger, et al. "Exploiting compositionality to explore a large space of model structures." UAI (2012).
[2] Duvenaud, David, et al. "Structure discovery in nonparametric regression through compositional kernel search." ICML (2013).

---

### Official Review · AnonReviewer3 · 2017-11-28
**Great results, needlessly overstated.**

**Rating:** 6
**Confidence:** 4

**Review:**

This work fits well into a growing body of research concerning the encoding of network topologies and training of topology via evolution or RL. The experimentation and basic results are probably sufficient for acceptance, but to this reviewer, the paper spins the actual experiments and results a too strongly.

The biggest two nitpicks:

> In our work we pursue an alternative approach: instead of restricting the search space directly, we allow the architectures to have flexible network topologies (arbitrary directed acyclic graphs)

This is a gross overstatement. The architectures considered in this paper are heavily restricted to be a stack of cells of uniform content interspersed with specifically and manually designed convolution, separable convolution, and pooling layers. Only the topology of the cells themselves are designed. The work is still great, but this misleading statement in the beginning of the paper left the rest of the paper with a dishonest aftertaste. As an exercise to the authors, count the hyperparameters used just to set up the learning problem in this paper and compare them to those used in describing the entire VGG-16 network. It seems fewer hyperparameters are needed to describe VGG-16, making this paper hardly an alternative to the "[common solution] to restrict the search space to reduce complexity and increase efficiency of architecture search."

> Table 1

Why is the second best method on CIFAR (“Hier. repr-n, random search (7000 samples)”) never tested on ImageNet? The omission is conspicuous. Just test it and report.

Smaller nitpicks:

> “New state of the art for evolutionary strategies on this task”

“Evolutionary Strategies”, at least as used in Salimans 2017, has a specific connotation of estimating and then following a gradient using random perturbations which this paper does not do. It may be more clear to change this phrase to “evolutionary methods” or similar.

> Our evolution algorithm is similar but more generic than the binary tournament selection (K = 2) used in a recent large-scale evolutionary method (Real et al., 2017).

A K=5% tournament does not seem more generic than a binary K=2 tournament. They’re just different.

---

### Official Review · AnonReviewer1 · 2017-11-30
**Good paper on searching space of network design**

**Rating:** 8
**Confidence:** 4

**Review:**

The authors present a novel evolution scheme applied to neural network architecture search. It relies on defining an expressive search space for conducting optimization, with a constrained search space that leads to a lighter and more efficient algorithm. To balance these constraints, they grow sub-modules in a hierarchical way to form more and more complex cells. Hence, each level is limited to a small search space while the system as a whole converges toward a complex structure. To speed up the search, they focus on finding cells instead of an entire network. In evaluation time, they insert these cells between layers of a network comparable in size to known networks. They find complex cells that lead to state-of-the-art performance on benchmark dataset CIFAR-10 and ImageNet. They also claim that their method is reaching a new milestone in evolutionary search strategies performance.

The method proposed for an hierarchical representation for optimizing over neural network designs is well thought and sound. It could lead to new insight on automating design of neural networks for given problems. In addition, the authors present results that appear to be on par with the state-of-the-art with architecture search on CIFAR-10 and ImageNet benchmark datasets. The paper presents a good work and is well articulated. However, it could benefit from additional details and a deeper analysis of the results.

The key idea is a smart evolution scheme. It circumvents the traditional tradeoff between search space size and complexity of the found models. The method is also appealing for its use of some kind of emergence between two levels of hierarchy. In fact, it could be argued that nature tends to exploit the same phenomenon when building more and more complex molecules. Thought, the paper could benefit from a more detailed analysis of the architectures found by the algorithm. Do the modules always become more complex as they jump from a level to another or there is some kind of inter-level redundancy? Are the cells found interpretable? The authors should try to give their opinion about the design obtained.

The implementation seems technically sound. The experiments and results section shows that the authors are confident and the evaluation seems correct. However, paragraphs on the architectures could be a bit clearer for the reader. The diagram could be more complete and reflect better the description. During evaluation, what is a step? A batch or an epoch or other?

The method seems relatively efficient as it took 36 hours to converge in a field traditionally considered as heavy in terms of computation, but at the requirement of using 200 GPU. It raises questions on the usability of the method for small labs. At some point, we will have to use insights from this search to stop early, when no improvement is expected. Also, authors claim that their method consume less computation time than reinforcement learning. This should be supported by some quantitative results.

The paper would greatly benefit from a deeper comparison over other techniques. For instance, it could describe more the advantages over reinforcement learning. An important contribution is to show that a well-defined architecture representation could lead to efficient cells with a simple randomized search. It could have taken more spaces in the paper.

I am also concerned the computational efficiency of the results obtained with this method on current processors. Indeed, the randomness of the found cells could be less efficient in terms of computation that what we can get from a well-structured network designed by hand. Exploiting the structure of the GPUs (cache size, sequential accesses, etc.) allows to get best possible performance from the hardware at hand. Does the solution obtained with the optimization can be run as efficiently? A short analysis forward pass time of optimized cells vs. popular models could be an interesting addition to the paper. This is a general comment over this kind of approach, but I think it should be addressed.

---

### Public Comment · ~Gabriel_Meyer-Lee1 · 2017-12-16
**Reproducibility of ICLR2018 Conference Paper893**

We implemented the deep neural network representation described in this paper as a part of the ICLR 2018 Reproducibility challenge and performed small-scale testing of the representation on the CIFAR-10 benchmark utilizing the described search methods.
Our implementation of the hierarchical encoding of a deep convolutional network was written in Python utilizing Keras with a TensorFlow backend. In the process of writing this implementation, we noticed several key omissions. We presumed that “depthwise” and “separable” convolutions refer to the definition in [Chollet 2017]. In this case, it would be impossible for the depthwise convolution operation to actually have a constant number of output filters as described. Furthermore, separable convolution would no longer be valid as a primitive as it could be produced by a stacked depthwise and 1x1 convolution. As such our implementation replaced depthwise with “standard” convolution. The described merging using depthwise concatenation requires padding the pooling operations, which is both unaddressed and contradicts the traditional use of pooling layers. Additionally, the identity operation is described as a primitive, but the initialization routine seems to imply that the identity operation is available at every level.
We also encountered several reproducibility issues in implementing the described evolutionary algorithm. The probability distributions used for random sampling in mutation are not given. We set all of them as uniform, but this biases the mutation method towards increasing complexity. This causes the number of random mutations per architecture in initialization to become an important but unknown parameter. We checked the number of parameters produced by generating 300 random hierarchical and flat architectures, first with 50 mutations each, then with 100 mutations each. The networks were assembled into the “small” CIFAR-10 architecture to check parameter numbers. The hierarchical architectures had 46 potential edges to mutate while the flat architecture had 55. The results of this showed that the flat architecture produced networks with 1.033 ± .574 M parameters after 50 mutations and 1.784 ± .889 M parameters after 100. The hierarchical architecture produced networks with .279 ± .168 M parameters after 50 mutations and .478 ± .280 M parameters after 100. These results show that random mutation does create a diverse initial population, but the complexity of that population is proportional to the number of mutations.
Despite all of the above mentioned issues, we were able to create a working implementation of the described system based solely off of the paper and so this submission must be given due credit as largely reproducible. Our small scale results do, however, indicate a few potential issues. We found a top validation fitness for random search on the hierarchical representation to be .73 with .42 M parameters and the top validation fitness for the flat representation to be .79 with 1.03 M parameters, both drawn from populations of 50. This is roughly in line with what’s shown in the figures at the top of section 4.2 of the submission, although the flat representation has far more parameters than any of the networks shown. The cause for this is likely due to above described omissions in the mutation and initiation routines.
The key obstacles to reproducing the results of this submission were the computational costs. The paper did clearly describe the costs of the experiments, but did not provide baseline results that could be replicated cheaply. The cheapest-to-compute reported results were the CIFAR-10 errors for randomly sampled architectures. Replicating these, however, is useless for evaluating the representation schemes in general or the search strategies. We did attempt this reproduction for about half the training time, producing inconclusive results of test accuracies of .79 for the hierarchical representation and .80 for the flat representation. Our recommendation, not just for these authors but for topology learning papers in general, is to augment the normal large scale benchmark-breaking experiments with mass small scale experiments. Ideally, an experiment could be run on a single GPU in one day. For this submission, this could be achieved by limiting training steps, evolution steps, or testing on an easier benchmark, like MNIST. The goal of these mass small scale experiments would be two fold: publishing results which are accessible for replication to a much larger population as well as conducting enough trials to demonstrate the statistical significance of the improvements shown by the paper’s novel methods. This would address a significant weak point of this paper, the indeterminate significance of the difference in performance between the flat and hierarchical representations.

References:

François Chollet. Xception: Deep learning with depthwise separable convolutions. arXiv preprint arXiv:1610.02357, 2017

---

> ### Author Response · Authors · 2018-01-05
> **Clarifications for reproducing the results**
>
> Thank you for taking the effort to implement our algorithm. Your detailed comments are valuable for us to improve the paper further. We provide the clarifications below, which would hopefully be useful for reproducing our results.
>
> * “it would be impossible for the depthwise convolution operation to actually have a constant number of output filters as described.”
> We did not require depthwise convolution operations to have a constant number of output filters (we’ll remove “of C channels” in the the 2nd bullet point in Sect. 2.3, which was a typo). This is not an issue because the way that we merge the nodes (depthwise concatenation) does not require the input nodes to have the same number of filters.
>
> * “separable convolution would no longer be valid as a primitive as it could be produced by a stacked depthwise and 1x1 convolution”
> In our case, each convolutional operation comes with batch normalization (BN) and ReLU, hence the separable convolution
> 3x3_depthwise_conv->1x1_conv->BN->ReLU
> is not exactly the same as the stack of
> 3x3_depthwise_conv->BN->ReLU and 1x1_conv->BN->ReLU.
> We also note that in general the algorithm remains valid if one primitive operation can be expressed in terms of the others.
>
> * “the initialization routine seems to imply that the identity operation is available at every level”
> No, the identity operation is only available at the motif level: a motif is initialised as a chain of identity operations, and a cell is initialised as a chain of motifs (note that a chain of identity chains is also an identity chain).
>
> * “The probability distributions used for random sampling in mutation are not given”
> We always use uniform distributions in all of our experiments. That being said, no hyperparameters were involved or tuned for mutation operations.
>
> * About the number of mutations during initialization.
> We would like to point out that a large number of mutations is necessary to produce a diverse initial population of architectures. In our case we used 1000.

---

### Author Response · Authors · 2018-01-05
**Responses to Reviewers**

We thank all reviewers for their comments. We will incorporate the suggested revisions into the new version of the paper. Our responses below focus on the major points.

* About comparing computation time with RL-based approaches (reviewer 1)
Our approach is faster than some published RL-based methods (e.g. 2000 GPU days in Zoph et al. (2017) vs 300 GPU days in our case). Having said that, we do not claim that evolution is more efficient than RL-based approaches in general.

* Efficiency of architectures found using architecture search (reviewer 1)
In terms of the number of parameters, our ImageNet model is comparable to Inception-Resnet-v2 but larger than NASNet-A. Although identifying fast/compact architectures was not the primary focus of this work, an interesting future direction is to include FLOPS or wall clock time as a part of the evolution fitness, letting the architecture search algorithm to discover architectures that are both accurate and computationally efficient.

* “During evaluation, what is a step?” (reviewer 1)
An evolution step refers to training and evaluation of a single architecture. We will make the definition more explicit in the revised paper.

* “The authors should try to give their opinion about the design obtained” (reviewer 1)
Our visualisation in appendix A shows that architecture search discovers a number of skip connections. For example, the cell contains a direct skip connection between input and output: nodes 1 and 5 are connected by Motif 4, which in turn contains a direct connection between input and output. The cell also contains several internal skip connections, through Motif 5 (which again comes with an input-to-output skip connection similar to Motif 4).

* “the paper spins the actual experiments and results a too strongly.” (reviewers 2 and 3)
Thank you for the suggested improvements. We will revise our writing and soften the claims.

* Missing ImageNet results for certain methods in Table 1 (reviewer 3)
ImageNet experiments under those two settings were still running at the time of the submission deadline. Their results are as follows:
Flat repr-n, parameter-constrained, evolution (7000 samples): 21.2 / 5.8
Hier. repr-n, random search (7000 samples): 21.0 / 5.5.
The latter result is due to the fact that the evolution fitness computed on CIFAR is a proxy for ImageNet performance. Computationally efficient architecture search directly on ImageNet is an interesting direction for future research.

* Mutation is biased towards adding edges (reviewer 2)
Indeed, in our implementation we don’t ensure an equal probability of adding and deleting edges. We think inferring the mutation bias along with evolution is an interesting direction for future work.

* Regarding a large number of hyperparameters specifying the architecture (reviewers 2 and 3)
We note that some hyperparameters can be adaptively tuned by evolution. Namely, M_l and |G^{(l)}| affect only the upper bounds on effective hyperparameters, since the algorithm may learn to not use a particular motif (hence the effective number of motifs becomes smaller than M_l), or to shortcut two nodes using an identity op (hence the effective number of nodes becomes smaller than |G^{(l)}|). Both behaviors have been empirically observed in our visualization (see Figure 5 & Figure 10 in Appendix A).

---

### Decision · Program_Chairs · 2018-01-29
**ICLR 2018 Conference Acceptance Decision**

**Decision:**

Accept (Poster)

**Comment:**

PROS:
1. Overall, the paper is well-written, clear in its exposition and technically sound.
2. With some caveats, an independent team concluded that the results were "largely reproducible"
3. The key idea is a smart evolution scheme. It circumvents the traditional tradeoff between search space size and complexity of the found models.
4. The implementation seems technically sound.

CONS:
1. The results were a bit over-stated (the authors promise to correct)
2. Could benefit from more comparison with other approaches (e.g. RL)